# Concentration of Macroelements and Trace Elements in Farmed Fallow Deer Antlers Depending on Age

**DOI:** 10.3390/ani12233409

**Published:** 2022-12-03

**Authors:** Katarzyna Tajchman, Aleksandra Ukalska-Jaruga, Francisco Ceacero, Monika Pecio, Żaneta Steiner-Bogdaszewska

**Affiliations:** 1Department of Animal Ethology and Wildlife Management, Faculty of Animal Sciences and Bioeconomy, University of Life Sciences in Lublin, Akademicka 13, 20-950 Lublin, Poland; 2Department of Soil Science Erosion and Land Protection, Institute of Soil Science and Plant Cultivation, State Research Institute, Czartoryskich 8, 24-100 Pulawy, Poland; 3Department of Animal Science and Food Processing, Czech University of Life Sciences Prague, Kamycka 129, 165 00 Praha-Suchdol, Czech Republic; 4Institute of Parasitology of the Polish Academy of Sciences, Research Station in Kosewo Górne, 11-700 Mrągowo, Poland

**Keywords:** *Dama dama*, availability of elements in antlers, diverse living conditions, ICP-MS, mineral composition of antlers, bone biology

## Abstract

**Simple Summary:**

The mineral content of the antler bone is one of the determinants of its mechanical strength, which directly affects success in competition with rivals. Therefore, this research aimed to analyze the concentration of five macroelements (Ca, P, Mg, K, Na) and nine trace elements (Li, V, Cr, Mn, Co, Cu, Zn, Se, Mo) in three characteristic antler sections, depending on the age of farmed fallow deer (*Dama dama*). The highest mean concentrations of macroelements (except K) were recorded in the youngest animals. With age and distance from the skull, Ca, P, Mg and Na contents decreased, while K increased. Higher mean concentrations of most trace elements (Cr, Mn, Co, Cu, Zn) were recorded in 3-year-old fallow deer in the distal antler positions. With an increase in the age, body mass and antler mass of farmed fallow deer, the concentration of Ca, P, Mg, K, Mn, Cu and Zn decreased in the studied tissue, whereas the Li concentration increased. There was a directly proportional relationship between the age and body mass of the animals, age and antler mass, and body mass and antler mass.

**Abstract:**

The mineral content of the antlers reflects the nutritional status and specific stage of bone growth in cervid males. Therefore, this research aimed to analyze the concentration of Ca, P, Mg, K, Na, Li, V, Cr, Mn, Co, Cu, Zn, Se and Mo in three characteristic antler positions selected based on the observation of fights between males. These were compared between farmed fallow deer (*Dama dama*) of different ages. The mineral compositions of tissues were analyzed using inductively coupled plasma mass spectrometry. The highest mean concentrations of macroelements (except K) were recorded in the youngest animals aged 2 or 3 years in the proximal position of the antlers. With age and distance from the skull, Ca, P, Mg and Na contents decreased, while K increased. Higher mean concentrations of most trace elements (Cr, Mn, Co, Cu, Zn) were recorded in 3-year-old animals in antler distal positions. With an increase in the age, body mass and antler mass of fallow deer, the concentration of Ca, P, Mg, K, Mn, Cu and Zn decreased (−0.414 ≤ R ≤ −0.737, *p* < 0.05) in the studied tissue, whereas Li increased (0.470 ≤ R ≤ 0.681, *p* < 0.05). The obtained results confirm that the antlers’ chemical composition changes with age, also changing the Ca:P ratio.

## 1. Introduction

Antlers have long been an exciting research material, especially in the context of basic bone biology research, because they are the only available animal tissue without the interference of surgical procedures and associated adverse effects [1]. An antler is one of the fastest-growing tissues of vertebrates [2,3] and accounts for 1 to 5% of the animal’s body weight [4], and even up to 6.3% of the young to 35% of the adult skeleton [2]. This rapid growth is associated with a high demand for minerals (mainly calcium) from food and partially extracted from the skeleton in a process called cyclic physiological osteoporosis [5,6,7,8,9,10,11]. In deer, the first antler traits are indicators of ontogenetic quality and may represent a phenotypic trait that reflects the nutritional status of the animal [12,13,14,15,16,17,18]. Mineral reserves accumulated in the animal’s body are depleted until the antlers grow, and physiological fatigue is visible precisely in the mineral composition, mechanical efficiency and porosity of the bony material of the antlers. Therefore, the mineral composition of antlers can be used as a diagnostic tool to assess macroelement and trace element deficiencies in deer [19]. In general, nutrition affects various bone characteristics, mainly mass or density [20,21,22,23], as well as microstructure [24] and mechanical properties [25]. However, in internal bones, such effects are often masked by remodeling processes [26], which is why it is better to study the impact of nutrition in antlers than in internal bones.

In farm breeding of cervids, it is possible to provide newborns and older animals with appropriate living conditions and well-balanced nutrition [15,27,28]. Many factors influence this physiological effort. Among them, nutrition is the most influential [1]. It is also an investment in the weapons that males are equipped with, and its relatively low mineral content is mainly responsible for their endurance in fights during the mating season [29,30], so it is desirable to have the largest possible antlers to reflect virility [31]. Moreover, since the antlers extend from the base or the first barb upwards, the successive side branches or the part forming the palmate may partially demineralize the skeleton and gradually deplete the accumulated reserves [32]; thus, the physiological effort should be reflected in a different composition or mechanical properties along the antler growth axis. The antlers are usually branched in structure, appearing to have evolved to interlock, allowing for the typical “thrust struggles” during the mating season [33,34]. During the clashes, the stags usually push each other against the beam and the tines on the palmate; therefore, these parts between the tines will be responsible for the result of a duel [35,36].

Antlers are only produced by males (except for the reindeer, *Rangifer tarandus*) and thus constitute a secondary sexual character. They are grown and cast from the tips of permanent frontal protuberances, known as pedicles [37,38]. This process is appositional and has been characterized as a special form of endochondral (and perichondral) ossification [39].

Most attention has been paid to red deer antlers (*Cervus elaphus*), although fallow deer are also an important trophy hunting and farmed species. Fallow deer antlers are usually shed in April and May, then new ones begin to grow almost immediately. In September, they are fully ossified and cleared of velvet. Unlike other deer, antlers in fallow deer are palmate in form and present a complex ornamental structure of spellers of various sizes [40].

The body weight and antler length of fallow deer who won or lost a fight were not correlated with fight duration, and nor, according to Jennings et al. [41], were the body weight of the heavier or lighter animal or the antler length of the animal that had longer or shorter antlers. Moreover, Jennings et al. [42] found no relationship between the presentation rate of the antler and antler size and symmetry. The internal structure of fallow deer antlers was analyzed [43,44,45], as was how the fallow deer’s antler histology changed after castration [46]. A positive effect of mineral supplementation [47] or amino acids on the growth, mass and concentration of minerals in the first antlers of fallow deer has also been demonstrated [48]. However, little attention has been paid to the analysis of the mineral composition of the antlers of older males of this species.

Therefore, the aim of our research was to assess the mineral composition of antlers at three key positions relative to the fighting technique of the species. Five macroelements (Ca—calcium, P—phosphorus, Mg—magnesium, K—potassium, Na—sodium) and nine trace elements (Li—lithium, V—vanadium, Cr—chromium, Mn—manganese, Co—cobalt, Cu—copper, Zn—zinc, Se—selenium, Mo—molybdenum), most of them with biological functions in the antler were studied. Farmed fallow deer (*Dama dama*) with different ages were used for comparing composition patterns at different stages of life.

## 2. Materials and Methods

### 2.1. Experimental Design

The analysis included 31 male deer 2–8 years old, divided into four groups: (I) 8 individuals, fourteen months old; (II) 6 three year olds; (III) 7 four–five year olds; group (IV) 10 six–eight year olds (Table 1). The farmed deer were bred at the Research Station of the Institute of Parasitology, Polish Academy of Sciences, Kosewo Górne in Poland. The breeding system was based on rotational pasture within plots with an area and density recommended by DEFRA [49], FEDFA [50] and Mattiello [51]. The study involved stags born naturally. In the early months, they were milk-fed by the doe, and later they fed on the vegetation available on the pasture. In the winter period (from December to March), the animals were fed ad libitum with grass haylage or hay with a moderate nutritional value. Each animal ingested on average 260 g d^−1^ of a mixture comprising 70% crushed oats, 15% rapeseed concentrate (containing 33% crude protein; Eko-pasz, Mońki, Poland), and 15% of soybean concentrate (with 45% crude protein content; Eko-pasz, Mońki, Poland) and Josera Phosphoreimer multi-ingredient licks (Josera, Nowy Tomyśl, Poland) (Appendix A).

### 2.2. Sampling

The body weight of the farmed animals was measured using MP 800 sensors coupled with a Tru-test DR 3000 weight reader (accuracy: 1%) standing inside a small handling box (2 m × 2 m × 0.6 m) with no need for sedation. Then, antlers from all animals were collected before the rutting period in 2020.

Antlers were cut about 1 cm above the burr for safety reasons once antlers were dead bone with no remaining velvet skin. The antlers were cut off with the animals physically immobilized in a handling box. The process was performed with a hand saw. This process is not painful for the animals since it is already a non-innervated dead bone. All animals were adapted to routine management and maintained good health and body condition during the experiment. This is a standard procedure performed every year in farmed male deer, preventing fights between bulls and thus, possible falls and injuries among the handlers.

The right beam from each antler was kept in a dry and ventilated room until the analyses when it was weighed with a precision balance WLC/C2/R (producer Radwag). Samples were collected at the same two positions along the main antler beam and the third from the middle of the palmate area to stabilize mean variability to allow us to study changes in mineral composition. Antlers were fragmented by a dental drill across the entire cross-section (cortical and cancellous layers together) from the whole first antler at three positions (between first and second tine, between second tine and palmate, and from the middle part of the palmate) from stags aged 3–8 years as in Figure 1.

### 2.3. Analysis of Macro- and Microelement Concentrations in Antlers

Antler samples analyzed for the content of macro- and trace elements were collected directly after slaughtering the animals and stored at room temperature until extraction. The lack of incineration of the samples to ash allowed the assessment of the current content of the analyzed elements in the dry matter of the antler without the risk of changes in its degree of oxidation. Additionally, the method of analysis used (in line with previously published works) allowed for the comparison of the results with other literature data. The analysis of macro- and microelement concentrations was conducted using inductively coupled plasma mass spectrometry (Agilent quadrupole 7500CE ICP-MS) following protocols widely used in the recent years for analyzing antler or bone samples [16,17,18]. The extracts were prepared in concentrated nitric acid by microwave digestion. A blank sample and certified reference material (NIST1400 and CRM028-050) were included in the analyses for quality control of the entire analytical process. The basic validation of the parameters included the recognition of recovery, ranging from 90 to 97%, and precision, defined as a relative standard deviation <3%. The limit of detection (LOD) was from 0.007 mg/kg to 0.099 mg/kg.

### 2.4. Statistical Analysis

The values of the analyzed variables are presented using the mean and standard deviation. The normality of the distribution of variables in the studied groups was checked using the Shapiro–Wilk normality test. The Kruskal–Wallis test was used to test the differences between the groups, and pairwise comparisons were made with the Mann–Whitney test with Bonferroni correction. The Spearman’s rank correlation test evaluated the relationship between body weight, antler weight, age and minerals. A significance level of *p* < 0.05 was adopted to indicate the existence of statistically significant differences or relationships. The statistical analyses were carried out using the Statistica 9.1 computer software (StatSoft, Poland).

## 3. Results

Table 2 and Table 3 show the average concentrations of macroelements and trace elements in farmed fallow deer antlers in four groups and positions 1, 2 and 3. The highest mean concentrations of macronutrients (except K) were recorded in the youngest animals, aged 2 or 3 years, in the first position (Table 2 and Table 3). With age and the distance from the skull, the content of Ca, P, Mg and Na decreased. Conversely, for the average K content in the antlers of the studied specimens, along with the distance from the skull, the concentration of this macronutrient increased in each group, but it also decreased with age. The highest mean Ca concentration was recorded in the antlers of young stags at 2 and 3 years of age (groups II and III, from 222.69 to 254.01 g/kg) in position 1, and the lowest in 6–8-year-old males (group IV) in position 3 (164.46 g/kg).

All the animals showed the highest mean Ca content in their antlers in the place closest to the skull (1st position); with increasing distance, the concentration of this element decreased. The highest mean concentration of P was recorded in the antlers of two-year-old individuals (122.44 g/kg), whereas relatively high but slightly lower levels were found in three-year-old fallow deer, and the lowest was found in the oldest stags in position 3 (76.19 g/kg). The highest mean Mg content was found in antlers in 2- and 3-year-old (group I and II) individuals in position 1 (4.95–4.89 g/kg) and the lowest in group III in position 3 (3.62 g/kg). The highest mean concentration of K was noted in 3-year-old individuals (group II) in position 3 (1.16 g/kg) and the lowest in animals from group III in position 1 (0.45 g/kg). In contrast, higher K values were observed in position 3 of the antlers compared to positions 1 and 2. The highest mean concentration of Na was found in 3-year-old individuals (group II) in antler position 1 (6.51 g/kg) and the lowest in stags from group III in position 3 (5.02 g/kg) (Table 2).

The Ca:P ratio, determining the efficiency of the skeleton, was the highest in group III in all antler positions (from 2.42 to 2.25) and slightly lower in group III (from 2.27 to 2.23), with the lowest being found in the youngest (group I, 1.94) and oldest (group IV, 2.15–2.16) animals (Table 2).

Higher mean concentrations of most trace elements (Cr, Mn, Co, Cu, Zn) were recorded in animals from group II in positions 2 and 3 of the antlers. The highest mean Li content was found in antler position 1 in individuals from group IV (0.25 mg/kg) and the lowest in fallow deer from group II in positions 2 and 3 (0.12 mg/kg in both cases). The highest mean concentration of V was recorded in group II animals in antler position 3 (0.08 mg/kg), and the lowest in group III and IV fallow deer in antler position 2 (0.01 mg/kg). The highest mean concentration of Cr was found in group II in antler position 2 (1.57 mg/kg) and the lowest in the youngest fallow deer (group I) (0.08 mg/kg; Table 2).

The highest mean amount of Mn was found in 3-year-old animals (group II) in position 2 in the antlers (9.32 mg/kg) and the lowest in the oldest animals (group IV) (2.64 mg/kg) in position 1. The highest mean content of Co was found in group II in position 2 (0.30 mg/kg), while the lowest was found in group III (0.08 mg/kg) in position 3. The highest average concentration of Cu was recorded in antlers in position 3 in 3-year-old fallow deer (group II, 1.34 mg/kg) and the lowest in the youngest cervids (group I, 0.37 mg/kg). The highest mean amount of Zn was recorded in young animals in group II in position 3 (64.81 mg/kg), while the lowest was recorded in group III in position 1 (40.93 mg/kg). High mean Se values were found in the oldest animals, with the highest in group III in the 3rd antler position (0.90 mg/kg) and the lowest in groups I and II in position 1 (0.05 mg/kg). The highest mean concentration of Mo was recorded in group II of fallow deer in position 1 in antlers (0.07 mg/kg) and the lowest in group II in position 1 (0.02 mg/kg) (Table 2).

The highest weight gain of farmed fallow deer was recorded between groups III and IV, i.e., 14.3 kg, and the lowest between groups I and II, i.e., 8.9 kg. The increase in antler mass was the highest between groups II and III (0.32 kg), and the lowest between groups I and II (0.09 kg); the oldest specimens (group IV, 0.13 kg) showed a slight increase in antler weight (Table 3).

A comparison of the mean concentrations of macroelements and trace elements from three positions between the three groups is presented in Table 2. A significant relationship was demonstrated between groups II and IV for all macronutrients (except Na) and also between groups II and III for P and K (*p* < 0.05). There was also a significant correlation between groups II and III for Li, V, Mn, Cu and Zn and between groups II and IV for Mn, Cu and Zn (*p* < 0.05). Moreover, there was a significant relationship between groups II and IV for animal body mass, antler mass and age (*p* < 0.05), between groups II and III for antler mass and between groups III and IV for fallow deer age (*p* < 0.05). A significant correlation between the age of the animals was also observed between groups II and IV, and III and IV (*p* < 0.001), body mass between groups II and IV (*p* = 0.002), and antler mass between groups II and III as well as II and IV (*p* = 0.001) (Table 3).

The measurements of macroelements and trace elements from individual places (positions 1, 2, 3) between groups of the studied animals were also analyzed (Table 4). A significant relationship was found between groups II–IV in each of the studied antler positions, as well as between groups I–IV in position 1 and groups II–III in position 3 for Ca content (*p* < 0.05). For the P concentration, a significant correlation was noted between groups I–III and I–IV in position 1, and between groups II–III and II–IV in positions 2 and 3 (*p* < 0.05). Moreover, a significant positive relationship was found between groups II and IV in antler position 2 for Mg concentration (*p* < 0.05), and between groups I–III in antler position 1 (*p* < 0.05) and between groups II–III and II–IV in positions 2 and 3 for K content (*p* < 0.05).

A significant relationship was also shown between groups II–IV in each antler position for Li concentration (*p* < 0.05), between groups I–IV in position 1, between groups II–III in positions 2 and 3, and between groups II–IV in position 2 for the concentration of V (*p* < 0.05). There was a significant relationship between groups II–III and II–IV in positions 2 and 3 for Mn (*p* < 0.05). In addition, a significant relationship was observed between groups II–IV in position 2 of the antlers for Cu concentration (*p* < 0.05), between groups I–IV as well as II–IV in position 1, and between groups II–III and II–IV in positions 2 and 3 for Zn concentration (*p* < 0.05). A significant age relationship was found between groups I–III, I–IV and II–IV in the first antler position, and between groups II–IV and III–IV in antler positions 2 and 3 (*p* <0.001). The body weight of animals also significantly correlated between groups I–III, I–IV and II–IV in antler position 1 (*p* <0.001), and in positions 2 and 3, only between groups II–IV (*p* = 0.002). There was a significant relationship with the weight of the antlers between groups I–III, and I–IV in position 1 (*p* <0.001), and between groups II–III and II–IV in positions 2 and 3 (*p* = 0.001) (Table 4).

To find out the relationship between the content of all the examined elements and the age of the animals, body mass and antler mass, the Spearman’s rank correlation coefficients were also calculated (Table 5). There was a significant negative relationship between all macroelements (except Na) with the age, body mass and antler mass of fallow deer stags (−0.438 ≤ R ≤ −0.734, *p* < 0.05). Some trace elements, Mn, Cu and Zn, were also negatively correlated (−0.414 ≤ R ≤ −0.737, *p* < 0.05) with age, body mass and antler mass, as was V with age (R = −0.541, *p* < 0.05). Only Li was significantly positively dependent on age, body mass and antler mass (0.470 ≤ R ≤ 0.681, *p* < 0.05). Moreover, a significant positive relationship was found between the age and body mass (R = 0.663, *p* < 0.05), between the age and antler mass (R = 0.665, *p* < 0.05) and between the body mass and antler mass of fallow deer (R = 0.727, *p* < 0.05) (Table 5).

## 4. Discussion

Our results show several interesting patterns regarding the concentration of macroelements. The Ca:P ratio was relatively stable across ages, between 2.15 and 2.42, except in Group I (just 1.94), but slightly higher than the recommended 2:1 ratio [52]. However, Ca and P contents always decreased along the main beam in Groups II, III and IV, following the same pattern described in other cervid species [5,11,53]. Moreover, the Ca and P contents were greatest at each studied position in Group II, then gradually decreased in Groups III and IV. In Group I, where just position 1 was sampled, the observed contents were slightly lower than in Group II. The good mineralization degree observed in 2- and 3-year-old animals (Groups I and II) may be related to the reserves accumulated in the skeleton during the lactation period, which affects first antler characteristics and may continue having an effect even up to 4 years of age [28]. Interestingly, the mineralization degree decreases with age, suggesting skeleton resources reduce with age. Adult fallow deer are known to be one of the deer species making the largest investment in antlers relative to body weight [54], which may explain this pattern. An alternative explanation would be that older animals have thicker antlers and/or thicker cortical bone, which may need less mineralized material to maintain good biomechanical properties. Both hypotheses can be supported by the negative correlation between Ca and P with age, body mass and antler weight. Further research including biomechanical analyses would be necessary to understand this pattern fully.

The same pattern already described for Ca and P was observed for Mg and Na, decreasing along the main beam and decreasing with age class. That is quite reasonable for Mg, which commonly replaces Ca in the hydroxyapatite crystals. Indeed, the same negative correlations with age, body mass and antler weight were also found for Mg. Conversely, this correlation was not found for Na, which is not an important element in bone biology and mainly reflects dietary levels [55] and blood-circulating Na content [56]. K shows the exact opposite pattern to Ca, P and Mg, as already described in red deer [5]. Previous studies have shown that the higher concentration of K in the distal portions reflects the greater amount of trabecular scaffold. Although the trabecular scaffold (or remnants of it) is found at all levels, it is in the distal parts where it is more abundant since the creation of primary osteons occurs directly on them [5,57,58]. However, this pattern was observed just in group II, indicating that, in general, the degree of mineralization was quite adequate in the antlers in this study (as expected in well cared for captive animals). An alternative hypothesis for the increase in K in the distal parts is that K reduces Ca losses via urine and supports antler development by mobilizing Ca from the skeleton [59]. Thus, K levels indicate physiological stress, which is higher in the distal parts of the antler when the available Ca is depleted [60].

All the studied macroelements showed values within the previously published ranges. For example, for the most important element (Ca), previous studies in fallow deer have reported both slightly lower [61] (in the same population) and higher Ca contents [46] in the first antler (Group I). Still, we should be cautious with these comparisons since content variations can be due to nutrition, soil mineral availability in the study area or even in the areas where the feedstuffs were grown. The comparison of absolute values with other cervid species is even more unclear since differences in mineral composition have been reported even in experimental settings with several deer species fed on the same diet during the antler growth period [11,62].

Among the trace elements, Mn, Cu and Zn are probably the most important ones for bone biology. Their contents did not show any clear patterns connected to the antler position. However, as in Grace et al. [53], they declined with age and increasing body and antler weight. Zn is especially relevant for making a correct interpretation of the results. Histochemistry studies have shown that Zn is found around incomplete osteons, thus indicating unfinished primary mineralization [5]. Increasing Zn content in the distal parts of the antler was not found in this study, suggesting a relatively complete process (even if secondary mineralization may still be unfinished [63]). This further supports the hypotheses previously discussed about the decrease in mineralization with age since that cannot be attributed to a lack of resources; for that to be true, the Zn values should have been higher in the distal parts compared to the proximal, which was not observed. Manganese seems to have a key role in creating the initial collagen matrix during the antler growth process [64], and deficiencies have been reported to negatively impact antlers’ mechanical properties, leading to increased antler fractures [17]. Manganese is poorly absorbed in ruminants, and its absorption may be reduced under high dietary calcium and phosphorus levels [1,65]. It seems, however, that these minerals did not interfere with Mn absorption in our research. Indeed, our results seem to agree with the described role of Mn since the highest levels were found in Group II, the one with greater mineralization (Ca and P). On the other hand, Cu content did not show any clear pattern, and no typical symptoms (hair depigmentation, lameness or exhaustion) of copper deficiency were observed.

Two aspects regarding Zn, Mn and Cu contents are worth further discussion. First, it is noticeable that these three important microminerals for bone biology are negatively correlated with age and body and antler weight, the same pattern previously described for the macrominerals with key roles in antler biology (Ca, P, Mg and K). Moreover, the concentration of the three of them (as well as V) was higher in 2- and 3-year-old fallow deer than in older ones; again, a similar pattern as previously described for most macroelements. This may also be related to increased reserves accumulated during lactation [28]. However, the skeleton is an important storage tissue just for Zn, while the other three are not easily stored in any tissue [64].

No visible pattern, differences connected to age class and position or correlation with age, body weight or antler weight were found for Se and Co, two trace elements with limited effects in bone biology. Selenium is relatively important, impairing bone metabolism and reducing bone density [66,67,68,69]. However, similarly to Cu, the levels found in antlers fall within the published ranges, and no clinical signs of Se deficiency were observed.

Other trace elements such as Li and Mo have a certain role in bone biology but have been scarcely studied in antlers. Lithium was the only element positively correlated with age, body weight and antler weight in this study. That may be due to the fact that Li can substitute Ca and Mg when these elements are scarce [15,16], which is supported by the fact that Li increased with age class (while Ca and Mg decreased). Mo is important in bone remodeling, which occurs very rarely in antlers. Probably for that reason (importance for bone biology but not for antler biology), no patterns, differences or correlations were found for Mo.

Finally, vanadium and chromium have not been studied in deer antlers. Vanadium seems to promote the formation of cartilage and bone tissue and participates in mineralizing bones and teeth [70]. Chromium helps prevent osteoarticular diseases such as osteoporosis and, as with K, limits the excessive excretion of calcium [71]. Just V showed significant results, but no clear pattern can be drawn, so further studies on this element are necessary.

## 5. Conclusions

This first exhaustive analysis of the mineral composition of fallow deer antlers confirms values and patterns similar to those previously observed in several red deer subspecies or Indian cervids. However, the continuous decrease in mineralization with age while showing no fatigue (no Zn or K increase in the top of the antler) or Ca deficiencies reflects the very high antler investment already well described for adult individuals of this species. The results also confirm the existence of numerous significant patterns and relationships (differences with age and position; correlations with age, body and antler weight) in those elements with key functions in antler biology (such as Ca, P, Mg, K, Zn, Mn, Se, Li or even V). In contrast, no clear and consistent patterns are observed for those minerals with low or no biological function (Na, Cu, Co or Cr). The observed patterns (more than the absolute values) may serve as a reference to understand future analyses of antlers from different origins and conditions and for detecting deficiencies in captive or wild settings since our results were obtained in captive animals raised under suitable conditions and producing antlers with no apparent deficiencies.

## Figures and Tables

**Figure 1 animals-12-03409-f001:**
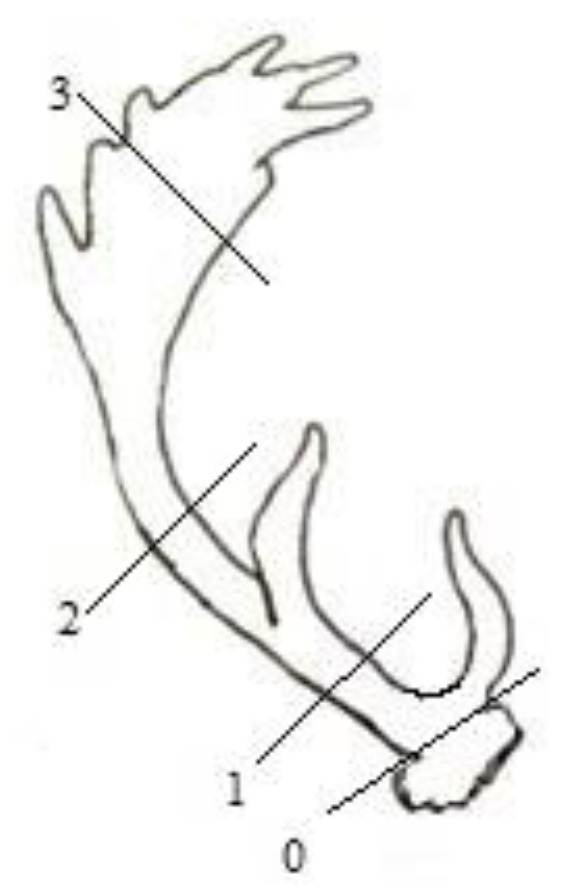
Places where samples 1–3 were obtained from individuals aged 3–8 years, 0—place where the antlers were cut.

**Table 1 animals-12-03409-t001:** Mean (M) and standard deviation (SD) age, body mass and antler mass of four groups of farmed fallow deer.

Analyzed Parameters		Group I (N = 8)		Group II (N = 6)		Group III (N = 7)		Group IV (N = 10)	
		M	SD	M	SD	M	SD	M	SD
Age	year	2.00	0.000	3.00	0.000	4.71	0.488	6.90	0.875
Body mass	kg	54.40	3.454	63.33	4.226	83.71	15.692	93.90	10.743
Antler mass	kg	0.02	0.004	0.29	0.045	0.70	0.111	0.76	0.136

**Table 2 animals-12-03409-t002:** Mean concentration (M) and standard deviation (SD) of macroelements and trace elements in the 1st, 2nd and 3rd positions of antlers (0.007 mg/kg < LOD < 0.099 mg/kg), and age, body mass and antler mass of farmed fallow deer.

Analyzed Parameters		Group I Position 1		Group II						Group III						Group IV					
				Position 1		Position 2		Position 3		Position 1		Position 2		Position 3		Position 1		Position 2		Position 3	
		M	SD	M	SD	M	SD	M	SD	M	SD	M	SD	M	SD	M	SD	M	SD	M	SD
Macroelements																					
Ca	g/kg	236.18	25.593	254.01	17.631	239.81	13.995	222.68	9.218	215.68	46.228	192.66	224.00	173.14	26.242	190.85	7.927	177.97	5.481	164.46	9.234
P		122.44	7.740	114.47	5.495	106.13	6.186	99.13	8.024	89.07	12.920	84.77	12.041	77.08	10.620	88.50	3.257	82.77	3.318	76.19	3.743
Mg		4.88	0.582	4.95	0.425	4.59	0.416	4.22	0.539	4.49	0.674	4.09	0.397	3.61	0.410	4.26	0.315	3.99	0.285	3.70	0.310
K		0.71	0.094	0.62	0.094	0.87	0.218	1.15	0.191	0.45	0.129	0.51	0.123	0.53	0.135	0.53	0.136	0.54	0.144	0.56	0.167
Na		6.17	0.655	6.51	0.673	5.76	0.601	5.03	0.735	6.18	1.255	5.58	0.597	5.01	0.678	5.72	0.377	5.45	0.362	5.09	0.356
Ca:P		1.94	0.262	2.23	0.223	2.27	0.212	2.26	0.204	2.42	0.371	2.28	0.102	2.25	0.103	2.16	0.091	2.15	0.063	2.16	0.119
Trace elements																					
Li	mg/kg	0.18	0.038	0.13	0.031	0.12	0.030	0.11	0.026	0.16	0.044	0.16	0.064	0.15	0.048	0.253	0.123	0.20	0.067	0.19	0.097
V		0.05	0.031	0.03	0.012	0.05	0.034	0.08	0.031	0.03	0.021	0.01	0.009	0.02	0.011	0.03	0.029	0.01	0.010	0.05	0.076
Cr		0.08	0.042	0.14	0.099	1.57	2.992	0.35	0.258	0.28	0.430	0.15	0.175	0.25	0.442	0.13	0.178	0.15	0.160	0.35	0.335
Mn		2.75	1.126	3.88	0.853	9.32	8.472	9.13	3.576	4.25	2.037	2.73	0.700	2.93	1.358	2.64	0.840	2.76	0.984	3.83	1.637
Co		0.13	0.058	0.14	0.035	0.30	0.259	0.16	0.067	0.21	0.284	0.13	0.085	0.07	0.026	0.13	0.180	0.12	0.074	0.19	0.204
Cu		0.37	0.241	0.56	0.085	1.19	0.741	1.34	0.932	0.54	0.558	0.41	0.122	0.46	0.176	0.43	0.370	0.44	0.301	0.68	0.516
Zn		59.27	11.314	58.82	8.432	61.71	11.416	64.81	8.158	47.85	12.948	43.18	5.502	41.96	5.404	40.93	5.158	42.54	8.581	44.11	5.225
Se		0.05	0.031	0.05	0.011	0.07	0.025	0.10	0.025	0.54	0.851	0.37	0.460	0.89	1.508	0.75	1.499	0.72	1.396	0.55	1.121
Mo		0.03	0.011	0.02	0.009	0.03	0.033	0.05	0.051	0.06	0.039	0.05	0.039	0.04	0.031	0.04	0.028	0.03	0.028	0.06	0.041

**Table 3 animals-12-03409-t003:** Comparison of the mean measurements from three sites between the three groups of animals.

Analyzed Parameters		Group II		Group III		Group IV		Kruskal–Wallis H Test (3, *N* = 23)	*p*	Correlation Coefficient between Groups
		M	SD	M	SD	M	SD			
Macroelements										
Ca	g/kg	238.84	10.301	193.83	28.943	177.76	5.635	11.398	0.003 *	II–IV, *p* = 0.002 *
P		106.58	5.273	83.64	11.741	82.49	2.572	10.843	0.004 *	II–III, *p* = 0.012 * II–IV, *p* = 0.008 *
Mg		4.59	0.426	4.07	0.399	3.98	0.271	7.081	0.029 *	II–IV, *p* = 0.041 *
K		0.88	0.155	0.50	0.118	0.55	0.103	12.929	0.002 *	II–III, *p* = 0.001 * II–IV, *p* = 0.018 *
Na		5.77	0.627	5.59	0.741	5.43	0.332	0.838	0.657	-
Ca:P		2.24	0.177	2.32	0.133	2.16	0.065	-	-	-
Trace elements										
Li	mg/kg	0.12	0.027	0.16	0.050	0.22	0.063	10.771	0.004 *	II–III, *p* = 0.003 *
V		0.06	0.015	0.02	0.012	0.03	0.027	7.584	0.023 *	II–III, *p* = 0.035 *
Cr		0.69	1.048	0.23	0.185	0.21	0.103	2.378	0.305	-
Mn		7.45	3.381	3.31	1.062	3.08	0.576	12.797	0.002 *	II–III, *p* = 0.010 * II–IV, *p* = 0.002 *
Co		0.20	0.074	0.14	0.112	0.15	0.087	2.875	0.237	-
Cu		1.03	0.448	0.47	0.256	0.52	0.173	10.006	0.006 *	II–III, *p* = 0.008 * II–IV, *p* = 0.032 *
Zn		61.78	8.415	44.33	7.432	42.53	4.984	11.536	0.003 *	II–III, *p* = 0.026 * II–IV, *p* = 0.003 *
Se		0.07	0.017	0.61	0.919	0.68	1.262	0.447	0.799	-
Mo		0.04	0.025	0.05	0.025	0.05	0.029	0.814	0.666	-
Measurements										
Age	year	3.00	0.000	4.71	0.488	6.90	0.875	19.946	<0.001 *	II–IV, *p* < 0.001 * III–IV, *p* = 0.033 *
Body mass	kg	63.33	4.226	83.71	15.692	93.90	10.743	13.005	0.002 *	II–IV, *p* = 0.001 *
Antler mass	kg	0.29	0.045	0.70	0.111	0.76	0.136	13.170	0.001 *	II–III, *p* = 0.019 * II–IV, *p* = 0.001 *

M—mean, SD—standard deviation, Kruskal–Wallis H test, * values statistically significant *p* < 0.05.

**Table 4 animals-12-03409-t004:** Comparison of measurements of macroelements and trace elements from individual positions between groups.

Analyzed Parameters	Position 1			Position 2			Position 3		
	Kruskal–Wallis H Test (3, *N* = 31)	*p*	Correlation Coefficient between Groups	Kruskal–Wallis H Test (3, *N* = 23)	*p*	Correlation Coefficient between Groups	Kruskal–Wallis H Test (3, *N* = 23)	*p*	Correlation Coefficient between Groups
Macroelements									
Ca	13.732	0.003 *	I–IV, *p* = 0.036 II–IV, *p* = 0.004	13.356	0.001 *	II–IV, *p* < 0.001	10.888	0.004 *	II–III, *p* = 0.022 II–IV, *p* = 0.005
*p*	21.587	0.0001 *	I–III, *p* < 0.001 I–IV, *p* < 0.001	10.378	0.005 *	II–III, *p* = 0.001 II–IV, *p* = 0.010	11.294	0.003 *	II–III, *p* = 0.012 II–IV, *p* = 0.006
Mg	7.377	0.061	-	6.855	0.032 *	II–IV, *p* = 0.034	4.987	0.083	-
K	11.938	0.007 *	I–III, *p* = 0.006	9.273	0.009 *	II–III, *p* = 0.014 II–IV, *p* = 0.028	12.015	0.002 *	II–III, *p* = 0.004 II–IV, *p* = 0.009
Na	4.401	0.221	-	0.628	0.730	-	1.007	0.605	-
Trace elements									
Li	11.241	0.011 *	II–IV, *p* = 0.006	7.795	0.020 *	II–IV, *p* = 0.016	8.212	0.016 *	II–IV, *p* = 0.012
V	8.753	0.033 *	I–IV, *p* = 0.025	8.389	0.015 *	II–III, *p* = 0.046 II–IV, *p* = 0.023	7.789	0.020 *	II–III, *p* = 0.025
Cr	3.146	0.369	-	2.401	0.301	-	2.634	0.268	-
Mn	8.417	0.038	-	11.311	0.003 *	II–III, *p* = 0.015 II–IV, *p* = 0.005	11.416	0.003 *	II–III, *p* = 0.003 II–IV, *p* = 0.031
Co	4.063	0.255	-	4.952	0.084	-	5.106	0.078	-
Cu	6.877	0.076	-	8.704	0.013 *	II–IV, *p* = 0.015	5.712	0.058	-
Zn	14.317	0.003 *	I–IV, *p* = 0.009 II–IV, *p* = 0.010	10.928	0.004 *	II–III, *p* = 0.024 II–IV, *p* = 0.004	12.882	0.002 *	II–III, *p* = 0.003 II–IV, *p* = 0.005
Se	6.268	0.099	-	0.594	0.742	-	1.282	0.527	-
Mo	6.681	0.083	-	0.764	0.683	-	1.238	0.539	-
Measurements									
Age	28.863	<0.001 *	I–III, *p* = 0.025 I–IV, *p* < 0.001 II–IV, *p* = 0.008	19.947	<0.001 *	II–IV, *p* < 0.001 III–IV, *p* = 0.033	19.947	<0.001 *	II–IV, *p* < 0.001 III–IV, *p* = 0.033
Body mass	24.496	<0.001 *	I–III, *p* = 0.003 I–IV, *p* < 0.001 II–IV, *p* = 0.045	13.005	0.002 *	II–IV, *p* = 0.001	13.005	0.002 *	II –IV, *p* = 0.001
Antler mass	24.686	<0.001 *	I–III, *p* = 0.001 I–IV, *p* < 0.001	13.170	0.001 *	II–III, *p* = 0.019 II–IV, *p* = 0.001	13.170	0.001 *	II–III, *p* = 0.019 II–IV, *p* = 0.001

Kruskal–Wallis H test, * values statistically significant *p* < 0.05.

**Table 5 animals-12-03409-t005:** Relationship between the concentration of macroelements and trace elements on age, body mass and antler mass of farmed fallow deer.

Analyzed Parameters N = 23	Age		Body Mass		Antler Mass	
	R	*p*	R	*p*	R	*p*
Macroelements						
Ca	−0.621	0.002 *	−0.582	0.003 *	−0.554	0.006 *
P	−0.571	0.004 *	−0.524	0.010 *	−0.515	0.012 *
Mg	−0.481	0.019 *	−0.477	0.021 *	−0.438	0.036 *
K	−0.481	0.019 *	−0.579	0.003 *	−0.734	<0.001 *
Na	−0.181	0.408	−0.325	0.129	−0.317	0.139
Trace elements						
Li	0.681	<0.001 *	0.608	0.002 *	0.470	0.024 *
V	−0.541	0.007 *	−0.177	0.416	−0.412	0.050
Cr	−0.322	0.134	−0.135	0.536	−0.362	0.088
Mn	−0.695	<0.001 *	−0.414	0.049 *	−0.661	<0.001 *
Co	−0.391	0.064	−0.109	0.618	−0.314	0.143
Cu	−0.535	0.008 *	−0.426	0.042 *	−0.569	0.004 *
Zn	−0.652	<0.001 *	−0.705	<0.001 *	−0.720	<0.001 *
Se	−0.028	0.897	0.175	0.423	0.229	0.291
Mo	−0.067	0.762	0.008	0.969	−0.077	0.726
Measurements						
Age animals	-	-	0.663	<0.001 *	0.665	<0.001 *
Body mass	0.663	<0.001 *	-	-		
Antler mass	0.665	<0.001 *	0.727	<0.001 *	-	-

R—Spearman rank correlation, * values statistically significant *p* < 0.05.

## Data Availability

Data are contained within the article or Appendix A.

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
