# Peer review of "Concentration of Macroelements and Trace Elements in Farmed Fallow Deer Antlers Depending on Age"

_animals, 2022, doi:10.3390/ani12233409_

Round 1
Reviewer 1 Report
In general, although there is a lot of well-planned work, this research has problems in three aspects:
1. Sampling.
……….“cortical and parenchymal layers together”, what exactly is meant by this?
In each section of antler positions 1, 2 and 3 it is not clear what part they have sampled for analysis: is it just cortical, or is it a mixture of cortical and cancellous bone?
If the analysis of any sample included a portion of cancellous bone, this is a problem since cancellous bone is less mineralized than cortical bone and is formed during antler growth much earlier than the primary osteones of cortical bone.
2. Methodology
The methodology used, although very sensitive, did not follow the usual steps for determining the chemical composition of bone mineral. This composition is determined on ashes of samples that are first dried at 100ºC in an oven, weighed and heated at 800ºC in an oven for 24h evaporating the organic matrix. It is essential to know the ash fraction in order to determine the concentration of elements in the ash. The ash fraction is an indicator (a proxy) of the mineralization degree, while the chemical composition, in particular the Ca/P ratio, determines the type of bone mineral. This mineral is an impure hydroxyapatite, variable, containing many structural substitutions (e.g., carbonates, fluorides, citrate and various trace elements).
In this work, the authors digest directly in nitric acid “natural” antler samples, i.e. samples containing an organic part and a mineral part. Although the authors have introduced controls in their analysis as certified reference material NIST1400 and CRM028-050 (for trace elements) this referee has doubts whether the results of the analysis of ash bone and natural bone samples are comparable. The method used has to be validated first. There are striking results, for example, the Ca/P ratio (2.15-2.16) of group IV is practically the same as the Ca/P ratio (2.12) of NIST1400. NIST1400 is ash bone from bovine bone burned at 1100°C, and is considered as a mature hydroxyapatite standard.
3. Meaning and interpretation of the results.
It is worth considering the following:
3.1. The chemical analysis only characterizes a CaP mineral, i.e., the measured Ca and P belong to a more or less mature hydroxyapatite mineral.
3.2. There is a different approach to the subject. During antler growth, the processes of osteogenesis (formation of bone tissue) and its mineralization (nucleation of a hydroxyapatite-type mineral) are necessarily incomplete because these (different) two processes have stopped with the shedding of the velvet which interrupts the blood flow, the supply of nutrients and also necroses the bone cells. In the antlers there is a marked proximal-distal spatiotemporal sequence, so histology and bone mineralization are different depending on the position. Furthermore, although osteogenesis is complete at a certain level (e.g., at position 1, the primary osteones have already completed their formation), mineralization is not.
Classically the mineralization process is described as occurring in two phases named primary mineralization and secondary mineralization. Primary mineralization is fast, days, reaching 70% of the mineralization degree. Secondary mineralization is slow, it takes months to reach 100% of the mineralization degree. Therefore mature hydroxyapatite (of higher crystallinity and with Ca/P ratio = 2.1) is not found in recently mineralized tissues. In other words, the mineral composition is the result of mineralization, which is always an unfinished process in the antler.
A recent study combining (spatially resolved) elemental analysis and histology in antler at positions 1, 2, and 3 can be found in:
Gomez, S., Garcia, A., Landete-Castillejos, T., Gallego, L., Pantelica, D., Pantelica, A., ... & Straticiuc, M. (2016). Potential of the Bucharest 3 MV Tandetron™ for IBA studies of deer antler mineralization. Nuclear Instruments and Methods in Physics Research Section B: Beam Interactions with Materials and Atoms, 371, 413-418.
3.3. Nutrition is an important factor in reaching peak bone mass (either in the skeleton or in the antler) but does not determine CaP-mineral composition in bone.
The authors can find an excellent review of the influence of nutrition in bone in the following book chapter:
Weaver, C.M and Gallant M.H. Chapter 14. Nutrition in Burr, D. B., & Allen, M. R. (Eds.). (2014). Basic and applied bone biology. Academic Press. pp 283-297.
Author Response
REVIEWER #1
In general, although there is a lot of well-planned work, this research has problems in three aspects:
- Sampling.
……….“cortical and parenchymal layers together”, what exactly is meant by this?
In each section of antler positions 1, 2 and 3 it is not clear what part they have sampled for analysis: is it just cortical, or is it a mixture of cortical and cancellous bone?
If the analysis of any sample included a portion of cancellous bone, this is a problem since cancellous bone is less mineralized than cortical bone and is formed during antler growth much earlier than the primary osteones of cortical bone.
Author´s response: Thanks for the comment. First, cancellous is used in the new draft instead of parenchymal. About the sampling method, the text is correct. We also agree with the reviewer that having studied just the cortical bone would have been better. However, the cortical layer is quite thin in the positions 2 and especially the position 3 in fallow deer making very difficult to sample just this part. Moreover, as the reviewer indicates, the volume and degree of mineralization of the cancellous bone is much smaller, so its influence in the results is relatively small. For these reasons we decided to sample cortical and cancellous bone together.
- Methodology
The methodology used, although very sensitive, did not follow the usual steps for determining the chemical composition of bone mineral. This composition is determined on ashes of samples that are first dried at 100ºC in an oven, weighed and heated at 800ºC in an oven for 24h evaporating the organic matrix. It is essential to know the ash fraction in order to determine the concentration of elements in the ash. The ash fraction is an indicator (a proxy) of the mineralization degree, while the chemical composition, in particular the Ca/P ratio, determines the type of bone mineral. This mineral is an impure hydroxyapatite, variable, containing many structural substitutions (e.g., carbonates, fluorides, citrate and various trace elements).
In this work, the authors digest directly in nitric acid “natural” antler samples, i.e. samples containing an organic part and a mineral part. Although the authors have introduced controls in their analysis as certified reference material NIST1400 and CRM028-050 (for trace elements) this referee has doubts whether the results of the analysis of ash bone and natural bone samples are comparable. The method used has to be validated first. There are striking results, for example, the Ca/P ratio (2.15-2.16) of group IV is practically the same as the Ca/P ratio (2.12) of NIST1400. NIST1400 is ash bone from bovine bone burned at 1100°C, and is considered as a mature hydroxyapatite standard.
Author´s response: The reviewer describes a methodology commonly used for analysing mineral composition in bone samples. The suggested methodology is especially adequate for analysing Ca and P. However, it would not be adequate for analysing certain trace elements like Se, which would disappear from the sample. Moreover, the methodology used has been widely used in recent years for analysing antler samples. Thus following the same methodology seems a better approach in order to get comparable data. References for this methodology were added to the new draft.
- Meaning and interpretation of the results.
It is worth considering the following:
3.1. The chemical analysis only characterizes a CaP mineral, i.e., the measured Ca and P belong to a more or less mature hydroxyapatite mineral.
Author´s response: We don´t understand this comment and how to use it for improving the manuscript. 14 minerals were studied, not just Ca and P. Indeed, as explained in the discussion, all the values obtained were within the published ranges for other deer species.
3.2. There is a different approach to the subject. During antler growth, the processes of osteogenesis (formation of bone tissue) and its mineralization (nucleation of a hydroxyapatite-type mineral) are necessarily incomplete because these (different) two processes have stopped with the shedding of the velvet which interrupts the blood flow, the supply of nutrients and also necroses the bone cells. In the antlers there is a marked proximal-distal spatiotemporal sequence, so histology and bone mineralization are different depending on the position. Furthermore, although osteogenesis is complete at a certain level (e.g., at position 1, the primary osteones have already completed their formation), mineralization is not.
Author´s response: Thanks for the comment. Indeed, the unfinished mineralization in the top compared to the base is what this study highlights.
Classically the mineralization process is described as occurring in two phases named primary mineralization and secondary mineralization. Primary mineralization is fast, days, reaching 70% of the mineralization degree. Secondary mineralization is slow, it takes months to reach 100% of the mineralization degree. Therefore mature hydroxyapatite (of higher crystallinity and with Ca/P ratio = 2.1) is not found in recently mineralized tissues. In other words, the mineral composition is the result of mineralization, which is always an unfinished process in the antler.
Author´s response: The author is correct. However, in antler biology it is common to find studies reporting Ca:P values ranging between 1.8 and 2.4, similarly to what we report in this study. A note about the unfinished secondary mineralization was added to the discussion.
A recent study combining (spatially resolved) elemental analysis and histology in antler at positions 1, 2, and 3 can be found in:
Gomez, S., Garcia, A., Landete-Castillejos, T., Gallego, L., Pantelica, D., Pantelica, A., ... & Straticiuc, M. (2016). Potential of the Bucharest 3 MV Tandetron™ for IBA studies of deer antler mineralization. Nuclear Instruments and Methods in Physics Research Section B: Beam Interactions with Materials and Atoms, 371, 413-418.
Author´s response: We are aware of this study. However, we didn´t use it in the new draft since we are not discussing histological aspects.
3.3. Nutrition is an important factor in reaching peak bone mass (either in the skeleton or in the antler) but does not determine CaP-mineral composition in bone.
The authors can find an excellent review of the influence of nutrition in bone in the following book chapter:
Weaver, C.M and Gallant M.H. Chapter 14. Nutrition in Burr, D. B., & Allen, M. R. (Eds.). (2014). Basic and applied bone biology. Academic Press. pp 283-297.
Author´s response: We are also aware about this review, which was also used in the new draft.

Reviewer 2 Report
This manuscript presents an extensive and detailed analysis of elements' composition of antlers of fallow deers. I believe these results are interesting and should be published, since they provide important information on nutrition and health of this species.
However, I have several concerns regarding this manuscript that I strongly suggest authors to improve.
1 - Abstract: The abstract structure does not follow "Instruction for Authors". Please, re-check these instructions. Abstracts should present "a single paragraph and should follow the style of structured abstracts, but without headings: 1) Background: Place the question addressed in a broad context and highlight the purpose of the study; 2) Methods: Describe briefly the main methods or treatments applied. Include any relevant preregistration numbers, and species and strains of any animals used. 3) Results: Summarize the article's main findings; and 4) Conclusion: Indicate the main conclusions or interpretations."
I believe You should not say at the abstract "in position 1 of the antlers" without providing any context about what position 1 is. In the method subsection of the abstract you should clearly state that measurements were made at three positions of the antlers (1,2 and 3).
2 - Introduction: The last paragraph of the introduction is confusing, presents a repetition of words (therefore, therefore) and I believe the last paragraph of any introduction should simply and clearly state the goal of your work and nothing more.
3- Methods: I understand it is a routine procedure in farmed fallow deers and to control injuries between males, but how exactly are the antlers cut? Do you use physical restrain techniques? How do you clean the remaining velvet skin? You say you don't use sedation to weight them, but what about to cut the antlers? All this should be specified in my opinion.
4- Results: Table 1 needs to be reconstructed. At first, it should be split into 2 tables from my perspective. The last section of the table where you present the age, body mass and antlers mass measurements (Measurements is not properly written) is hard to read. The means and SD are referred to groups and not to positions, of course, but it becomes hard to follow the columns. Moreover, the column titles should not the "1 position, 2 positions...." but " "position 1, position 2", as you actually present in the whole manuscript.
The Results section is too long in my opinion, perhaps some information could not be presented in the text because it is already presented with the tables. Just highlight the most relevant results and the ones you are going to discuss more in the Discussion.
5- Discussion: Paragraph 3 - there is no reference to the first sentence of this paragraph regarding the soil composition and differences between Spain and Poland and I believe that is very relevant. More than just comparing your study with others (as your largely and properly do) I think that you should also try to explain your findings a little bit more than you do. Furthermore, what are the limitations of your methodology? This should also be more extensively discussed
6- Conclusion - You simply summarise your most relevant results. You should also provide future perspectives about similar research or management improvements you think you should apply to this population. What do all these results tell you about these fallow deers' health, nutrition or management after all?
Author Response
REVIEWER #2
This manuscript presents an extensive and detailed analysis of elements' composition of antlers of fallow deers. I believe these results are interesting and should be published, since they provide important information on nutrition and health of this species.
However, I have several concerns regarding this manuscript that I strongly suggest authors to improve.
1 - Abstract: The abstract structure does not follow "Instruction for Authors". Please, re-check these instructions. Abstracts should present "a single paragraph and should follow the style of structured abstracts, but without headings: 1) Background: Place the question addressed in a broad context and highlight the purpose of the study; 2) Methods: Describe briefly the main methods or treatments applied. Include any relevant preregistration numbers, and species and strains of any animals used. 3) Results: Summarize the article's main findings; and 4) Conclusion: Indicate the main conclusions or interpretations."
Author´s response: The abstract was modified following the advices.
I believe You should not say at the abstract "in position 1 of the antlers" without providing any context about what position 1 is. In the method subsection of the abstract you should clearly state that measurements were made at three positions of the antlers (1,2 and 3).
Author´s response: A different terminology was used in the abstract, as suggested by the reviewer.
2 - Introduction: The last paragraph of the introduction is confusing, presents a repetition of words (therefore, therefore) and I believe the last paragraph of any introduction should simply and clearly state the goal of your work and nothing more.
Author´s response: Changed as suggested in the new draft.
3- Methods: I understand it is a routine procedure in farmed fallow deers and to control injuries between males, but how exactly are the antlers cut? Do you use physical restrain techniques? How do you clean the remaining velvet skin? You say you don't use sedation to weight them, but what about to cut the antlers? All this should be specified in my opinion.
Author´s response: Further details about this procedure were added to the new draft.
4- Results: Table 1 needs to be reconstructed. At first, it should be split into 2 tables from my perspective. The last section of the table where you present the age, body mass and antlers mass measurements (Measurements is not properly written) is hard to read. The means and SD are referred to groups and not to positions, of course, but it becomes hard to follow the columns. Moreover, the column titles should not the "1 position, 2 positions...." but " "position 1, position 2", as you actually present in the whole manuscript.
Author´s response: The suggested changes were implemented in the new draft.
The Results section is too long in my opinion, perhaps some information could not be presented in the text because it is already presented with the tables. Just highlight the most relevant results and the ones you are going to discuss more in the Discussion.
Author´s response: We understand this comment. The section has been slightly reduced in the new draft.
5- Discussion: Paragraph 3 - there is no reference to the first sentence of this paragraph regarding the soil composition and differences between Spain and Poland and I believe that is very relevant. More than just comparing your study with others (as your largely and properly do) I think that you should also try to explain your findings a little bit more than you do. Furthermore, what are the limitations of your methodology? This should also be more extensively discussed
Author´s response: The discussion has been deeply remodelled in this new draft, and the suggestions from the reviewer were implemented.
Please, note that the Discussion is not changes-tracked in the new draft because the rection has been fully restructured.
6- Conclusion - You simply summarise your most relevant results. You should also provide future perspectives about similar research or management improvements you think you should apply to this population. What do all these results tell you about these fallow deers' health, nutrition or management after all?
Author´s response: The conclusion section has been fully rewritten in the new draft.

Reviewer 3 Report
The purpose of the reviewed work was to analyze concentration of macroelements and trace elements in farmed fallow deer antlers depending on age. The results of the analysis were collected from 31 male deer, were bred at the Research Station of the Institute of Parasitology, Polish Academy of Sciences, Kosewo Górne in Poland. The subject of article is adequate to its content. The article is written in a clear and understandable way. The authors have contributed a lot of work, but have not avoided some shortcomings.
Recomendation 1: In the introduction [page 2 (36,37)], it is unnecessary to quote the same thing twice. These quotations can be inserted after the sentence: "Moreover, they are grown and cast from the tips of permanent frontal protuberances, known as pedicles.”
Recomendation 2: The beginning of the discussion should be rewritten. Most of the information contained there is more suitable for results.
Recomendation 3: The final conclusions should be slightly modified, as they are an abbreviated rearrangement of the results.
Recomendation 3: In the conclusions, the authors should demonstrate the importance of analyzing the concentration of macronutrients and trace elements in fallow deer antlers
The comments do not diminish the value of the paper, and I think that with the recommended corrections, the manuscript may be ready for publication
Author Response
REVIEWER #3
The purpose of the reviewed work was to analyze concentration of macroelements and trace elements in farmed fallow deer antlers depending on age. The results of the analysis were collected from 31 male deer, were bred at the Research Station of the Institute of Parasitology, Polish Academy of Sciences, Kosewo Górne in Poland. The subject of article is adequate to its content. The article is written in a clear and understandable way. The authors have contributed a lot of work, but have not avoided some shortcomings.
Recomendation 1: In the introduction [page 2 (36,37)], it is unnecessary to quote the same thing twice. These quotations can be inserted after the sentence: "Moreover, they are grown and cast from the tips of permanent frontal protuberances, known as pedicles.”
Author´s response: Changed as suggested.
Recomendation 2: The beginning of the discussion should be rewritten. Most of the information contained there is more suitable for results.
Author´s response: The whole discussion has been extensively rewritten.
Recomendation 3: The final conclusions should be slightly modified, as they are an abbreviated rearrangement of the results.
Recomendation 3: In the conclusions, the authors should demonstrate the importance of analyzing the concentration of macronutrients and trace elements in fallow deer antlers
Author´s response: The conclusions have been rewritten, as also suggested by reviwer 2.
The comments do not diminish the value of the paper, and I think that with the recommended corrections, the manuscript may be ready for publication
Round 2
Reviewer 1 Report
The second version did not clarify some points related to the reviewer's comments. Two sections still need to be revised.
1. Methodology
2.3. Analysis of macro- and microelements concentrations in antlers (Lines 148-157)
The authors still do not clarify the condition of the samples for analysis:
Are they (room)-dry samples (directly digested)?
Are they (oven)-dry samples (directly digested)?
Are they ashed samples (directly digested)?
The authors now cite other papers (16-18) and say that they followed the stated methodology. However, in these papers, the analysis was performed on ashed samples.
This point causes further confusion. If the samples analyzed are not ash samples, they should validate their method by comparing their results to other ash samples obtained from the same site. If the samples analyzed are ash samples, they should report the ash fraction (%) and the values for Ca, Mg, and P as percentages.
This point is very important. The results in Table 2 are particularly striking in Group IV.
Their results would be of great value if compared and confirmed with the ash samples.
On the other hand, for each element analyzed, it is necessary to indicate what the limit of detection (LOD) was. (This limit could be included in Table 2 for clarity).
2. Discussion
Everything is easier to understand if the chemical composition, the mineral content, is related to the growth stage of the antler, especially to the proportion of bone tissue types present at each level of the antler and their degree of mineralization. This is due to the fact that during the formation of an antler, from the base to the tip, different stages of the growth process can be observed simultaneously. A very detailed dynamic study of the different stages can be found in:
S. Gomez, A.J. García, S. Luna, U. Kierdorf, H. Kierdorf, L. Gallego, T. Landete-Castillejos, Labeling studies on cortical bone formation in the antlers of red deer (Cervus elaphus), Bone 52 (2013) 506-515.
With this approach more things are explained more simply (without recurring to vague concepts such as “physiological fatigue”). For example, the higher concentration of K in the distal portions is explained by the greater amount of trabecular scaffold in these. The study (5) performed K-histochemistry and showed that the highest K concentration was located precisely at the level of the trabecular scaffold. Although the trabecular scaffold (or remnants of it) is found at all levels, it is in the distal parts where it is more abundant. In these places there has been no resorption and the primary osteones form directly on them.
The article:
S. Gomez, A. Garcia, T. Landete-Castillejos, L. Gallego, D. Pantelica, A. Pantelica, A. Pantelica, M. Straticiuc, Potential of the Bucharest 3 MV Tandetron™ for IBA studies of deer antler mineralization, Nucl. Instr. Meth. Phys. Res. B 371 (2016) 413-418.
should be quoted. It is not about histology but about the measurement of elements in three antler positions with a very sensitive method such as PIXE.
Further comments.
Line 16-17:
The mineral composition of the bone tissue of antlers is responsible for its mechanical resistance…..
The phrase is not accurate. The reviewer suggests changing it.
The mineral content of the antler bone is one of the determinants of its mechanical strength,
(Because there are other determinants besides mineral content, such as apparent mineral density, porosity, and microstructure. The antler is an elastic, anisotropic, heterogeneous and composite material).
Lines 28-29
The mineral content of the antlers reflects the nutritional status and level of physiological fatigue in cervids males…
This reviewer suggests changing it.
……………………….and level of physiological processes of bone growth in cervid males.
Or
……………………. and specific stage of bone growth in cervid males.
Line 50
This rapid growth is associated with a high demand for minerals extracted from the skeleton in a process called cyclic physiological osteoporosis [6-11].
It should be noted that the mineral (mainly calcium) is also obtained from food and only partially from the skeleton.
Moreover, the cited authors speak of a high bone turnover, implying a process of bone remodeling (resorption followed by formation). In other words: During the growth of the antlers, new bone tissue is also formed in the skeleton at the same time!
Line 65-66
and its mineral composition is mainly responsible for their endurance in fights during the mating season
This reviewer suggests to specify :
…its low mineral content……………
(The most outstanding mechanical properties of antler compared to bone are the higher values for Work to peak force (W) and Impact energy absorption (U); these are due to the low degree of mineralization and the collagenous fibrous structure.)
Author Response
Reviewer#1
The second version did not clarify some points related to the reviewer's comments. Two sections still need to be revised.
- Methodology
2.3. Analysis of macro- and microelements concentrations in antlers (Lines 148-157). The authors still do not clarify the condition of the samples for analysis: Are they (room)-dry samples (directly digested)? Are they (oven)-dry samples (directly digested)? Are they ashed samples (directly digested)? The authors now cite other papers (16-18) and say that they followed the stated methodology. However, in these papers, the analysis was performed on ashed samples. This point causes further confusion. If the samples analyzed are not ash samples, they should validate their method by comparing their results to other ash samples obtained from the same site. If the samples analyzed are ash samples, they should report the ash fraction (%) and the values for Ca, Mg, and P as percentages. This point is very important. The results in Table 2 are particularly striking in Group IV. Their results would be of great value if compared and confirmed with the ash samples.
Author´s response: The samples used in this study were room-dry samples with direct digestion. We apologize for the mistake with the references due to the different changes in the manuscript. The references are now correct in the new draft. Finally, the results for macro-elements are shown as g/kg, which is also a very normal way of presenting this information. But we are open to show it as percentage if suggested by the editor.
On the other hand, for each element analyzed, it is necessary to indicate what the limit of detection (LOD) was. (This limit could be included in Table 2 for clarity).
Author´s response: Maximum and minimum LOD were added in methods section in the new draft.
- Discussion
Everything is easier to understand if the chemical composition, the mineral content, is related to the growth stage of the antler, especially to the proportion of bone tissue types present at each level of the antler and their degree of mineralization. This is due to the fact that during the formation of an antler, from the base to the tip, different stages of the growth process can be observed simultaneously. A very detailed dynamic study of the different stages can be found in:
- Gomez, A.J. García, S. Luna, U. Kierdorf, H. Kierdorf, L. Gallego, T. Landete-Castillejos, Labeling studies on cortical bone formation in the antlers of red deer (Cervus elaphus), Bone 52 (2013) 506-515.
With this approach more things are explained more simply (without recurring to vague concepts such as “physiological fatigue”). For example, the higher concentration of K in the distal portions is explained by the greater amount of trabecular scaffold in these. The study (5) performed K-histochemistry and showed that the highest K concentration was located precisely at the level of the trabecular scaffold. Although the trabecular scaffold (or remnants of it) is found at all levels, it is in the distal parts where it is more abundant. In these places there has been no resorption and the primary osteones form directly on them.
Author´s response: We are aware of this study and the explanation provided for the increase in K in the distal parts. This was included in the new draft. Please, notice that the K level was indeed quite similar in all the studied groups, with the described increasing pattern occurring just in Group II. Together with the similar result obtained for Zn (another mineral indicating “incomplete” mineralization) the results suggest that the antlers were in general quite well formed at all levels, which is an essential point for the interpretation of the results.
The article: S. Gomez, A. Garcia, T. Landete-Castillejos, L. Gallego, D. Pantelica, A. Pantelica, A. Pantelica, M. Straticiuc, Potential of the Bucharest 3 MV Tandetron™ for IBA studies of deer antler mineralization, Nucl. Instr. Meth. Phys. Res. B 371 (2016) 413-418. should be quoted. It is not about histology but about the measurement of elements in three antler positions with a very sensitive method such as PIXE.
Author´s response: As indicated in the previous answer, this reference has been included in the new draft.
Further comments.
Line 16-17: The mineral composition of the bone tissue of antlers is responsible for its mechanical resistance….. The phrase is not accurate. The reviewer suggests changing it. The mineral content of the antler bone is one of the determinants of its mechanical strength, (Because there are other determinants besides mineral content, such as apparent mineral density, porosity, and microstructure. The antler is an elastic, anisotropic, heterogeneous and composite material).
Author´s response: Correct. The sentence has been reworded accordingly.
Lines 28-29: The mineral content of the antlers reflects the nutritional status and level of physiological fatigue in cervids males… This reviewer suggests changing it. ……………………….and level of physiological processes of bone growth in cervid males. Or ……………………. and specific stage of bone growth in cervid males.
Author´s response: We also agree and the sentence has been also reworded following the suggestion.
Line 50: This rapid growth is associated with a high demand for minerals extracted from the skeleton in a process called cyclic physiological osteoporosis [6-11]. It should be noted that the mineral (mainly calcium) is also obtained from food and only partially from the skeleton. Moreover, the cited authors speak of a high bone turnover, implying a process of bone remodeling 2also formed in the skeleton at the same time!
Author´s response: We reworded the sentence according to the comment. It is already widely accepted after the studies by Banks that the mineral resourced supporting antler growth are coming from both sources, quite in a 50-50% basis.
Line 65-66: and its mineral composition is mainly responsible for their endurance in fights during the mating season. This reviewer suggests to specify : …its low mineral content…………… (The most outstanding mechanical properties of antler compared to bone are the higher values for Work to peak force (W) and Impact energy absorption (U); these are due to the low degree of mineralization and the collagenous fibrous structure.)
Author´s response: The comment is correct and the sentence has been reworded according to it.

Reviewer 2 Report
Authors considerably modified their manuscript according to the comments, corrections, and suggestions provided in my first review report, significantly improving their manuscript.
However, when they are asked to answer point-by-point to the corrections they should literally tell me what they did in the text to improve it and not simply answer "changes have been made" to almost all the points. This is not usually the point of making a poin-by-point answer to reviewers' comments. Nevertheless, I was able to see where the changes were made. The abstract was modified. Tables are clearly more understandable, methodology provides more details about how the procedures were taken. However, I still find the last paragraph of the introduction insufficiently clear about the aims of the study. I would recommend having a single paragraph specifying what evaluations were made and what are the goals of the study.
Moreover, conclusions were improved, but there are sentences that need to be revised because in this form they are confusing:
"The observed patterns (more than the absolute values) observed in captive animals raised under suitable conditions and growing antlers with no apparent deficiencies may serve as..."
Author Response
Authors considerably modified their manuscript according to the comments, corrections, and suggestions provided in my first review report, significantly improving their manuscript.
However, when they are asked to answer point-by-point to the corrections they should literally tell me what they did in the text to improve it and not simply answer "changes have been made" to almost all the points. This is not usually the point of making a poin-by-point answer to reviewers' comments. Nevertheless, I was able to see where the changes were made. The abstract was modified. Tables are clearly more understandable, methodology provides more details about how the procedures were taken. However, I still find the last paragraph of the introduction insufficiently clear about the aims of the study. I would recommend having a single paragraph specifying what evaluations were made and what are the goals of the study.
Author´s response: We apologize for not having detailed in the previous response letter all the changes done in the previous draft. In the new draft the last paragraph of the introduction has been modified (L98-104 in the new clean draft).
Moreover, conclusions were improved, but there are sentences that need to be revised because in this form they are confusing: "The observed patterns (more than the absolute values) observed in captive animals raised under suitable conditions and growing antlers with no apparent deficiencies may serve as..."
Author´s response: We agree that some sentences in the conclusion were not well written in the previous draft. They have been clarified in the new one (L385-398 in the new clean draft).
